# Prevalence of Low Back Pain among Primary School Students from the City of Valencia (Spain)

**DOI:** 10.3390/healthcare9030270

**Published:** 2021-03-03

**Authors:** Vicente Miñana-Signes, Manuel Monfort-Pañego, Antonio Hans Bosh-Bivià, Matias Noll

**Affiliations:** 1Academic Unit of Physical Education, Body Languages Didactics Department, Teacher Training Faculty, University of Valencia, 46022 Valencia, Spain; manuel.monfort@uv.es (M.M.-P.); Antonio.Bosch@uv.es (A.H.B.-B.); 2Public Health Department, Instituto Federal Goiano, Ceres 76300-000, Goias, Brazil; matias.noll@ifgoiano.edu.br; 3Department of Sports Science and Clinical Biomechanics, University of Southern Denmark, 5230 Odense, Denmark

**Keywords:** back health, low back pain, prevalence, kindergarten, primary school, children

## Abstract

It is well-known that low back pain (LBP) prevalence is high among school-age children. However, literature concerning the initial onset of back pain between the ages of three and eleven years is scarce. The present study aims to analyze the prevalence of LBP in kindergarten and primary school students. A total of 278 (9.9 ± 2.1 years old; 52.2% girls) students from two public kindergartens and primary schools in Valencia (Spain) participated in this cross-sectional study. The Nordic questionnaire on LBP was used to assess the onset and duration of LBP symptoms. The lifetime prevalence of LBP was 47.5% (*n* = 132), the last year’s prevalence was 44.2% (*n* = 123), and last week’s prevalence was 18.8% (*n* = 50). Boys and girls reported a lifetime prevalence of 52.3% (*n* = 64) and 47.7% (*n* = 63) (*p* = 0.186, Fisher’s exact test, 2-sided), respectively. By age group, lifetime episodes of LBP became more prevalent with increasing age (*p* < 0.001, Fisher’s exact test, 2-sided). In summary, our findings show that LBP increases with age and further strengthens the evidence that LBP onset could start as young as 10 years of age.

## 1. Introduction

It is well known that low back pain (LBP) prevalence is high throughout school-aged children’s lives [1,2]. Moreover, LBP is an important health problem worldwide [3,4,5] (Figure 1). However, literature investigating the initial onset of LBP in children between six and eleven years old is scarce.

The latest study in Spain used a cohort of 1500 adolescents from the Valencian Community between the ages of twelve and eighteen and detected a 44.5% lifetime prevalence of LBP. Prevalence was higher in girls (50.3%) than in boys (38.9%) and reached 36.9% by age of thirteen [6]. In addition, a growing body of evidence indicates that LBP onset occurs between ten and fourteen years of age [4,7,8], with prevalence increasing with age and approaching adult levels around eighteen years of age [9].

Acute LBP is a symptom rather than a disease [10]. The condition is common, and it is usually not possible to diagnose a specific pathoanatomical cause [11]. Like other symptoms, such as headache or dizziness, LBP may cause significant discomfort and impairment in a child’s everyday life and contribute to long-term health problems. Research suggests that children and adolescents reporting back pain require increased healthcare utilization and report absenteeism, impairments at school, restricted physical activity, or combinations of these adversities more often [12].

Randomized control trials on back health education in school settings demonstrate that educational interventions effectively improve awareness, postural habits, core muscle endurance, and back health in children [13,14,15,16,17,18]. Along this line and to facilitate the evaluation of educational interventions, other studies [19,20,21,22] have focused on elaborating and validating the design of reliable evaluation instruments to assess contents related to back health. The contents addressed by these studies include awareness, postural habits, and back health in the school setting. However, these works focused on the prepuberal and adolescent population only. Therefore, studies on young primary school students are scarce and necessary to improve back health through early detection and interventions.

Given the present context and due to the lack of literature on LBP, our study aims to analyze the prevalence of LBP in students during primary education to describe and assess the situation and promote further studies focusing on that stage.

## 2. Materials and Methods

Cross-sectional study design.

### 2.1. Participants

The study was performed in Valencia (Spain), with an eligible population of 40,890 primary education students during the 2019–2020 school year. This age group was selected because non-specific LBP onset occurs in children aged 10–14 years old according to literature [6,8], and we wanted to describe the situation in a younger age group. The initial sample consisted of a total of 566 students chosen by convenience from two public primary schools (school 1 and school 2) in the city of Valencia, Spain. Of the 566 invited students, 264 (46.6%; 9.4 ± 1.3 years old; 52.7% girls, *n* = 139) children completed the questionnaires (Table 1) and were included in the present study. 

The sample comprised a representative primary education student cohort with a 95% confidence interval (CI) and a margin of error of ± 6%. We observed that 32 (12.1%) were in the 1st cycle of primary education, 89 (33.7%) in second cycle, and 143 (54.2%) in third cycle.

### 2.2. Selection Criteria

The inclusion criteria were as follows: students must be aged between six and eleven years of age and attending primary school (one of the two selected schools in Valencia) during the study period. Students must have also had sufficient capacity to understand the questionnaire and complete it. The youngest students (6–7 years old) completed the questionnaire with the help of an adult.

Exclusion criteria consisted of the following: students who did not return the informed consent form signed by their parents or guardians, those who did not participate due to illness or disability, and those who did not complete all questionnaire items.

### 2.3. Evaluation Instruments

An adapted version of the Nordic questionnaire on LBP [23] was used. The document included questions regarding the duration of LBP symptoms over time and was handed out to all students, although it has not been validated for the children population. Sociodemographic data included sex and age.

### 2.4. Procedure

The questionnaire was administered at school or home. Classroom teachers gave away the questionnaire at the school’s computer room or provided families with the guide to complete it. The questionnaire was available on Google Forms.

School-age students have been more exposed to current technology throughout their lives than older individuals, which might make it more likely for them to return surveys either in paper-and-pencil or computer formats [24]. As in other studies [18,25], the questionnaires were completed through self-registration by the students and with the help of their parents in the case of the youngest children (6–8 year old). The latter were required to answer only dichotomic (Yes or No) answers about LBP. 

All students voluntarily participated in the study. The school administrators, class tutors, and parents were informed about the study in writing and expressed their consent. The study was approved by the Ethics Committee in experimental research at the University of Valencia (reference number: H1509086047576).

### 2.5. Statistical Analysis

Descriptive statistics, including means, standard deviation, frequencies, and percentages, were obtained to represent LBP prevalence data between sex and age groups. Three types of analyses were considered to assess the prevalence rate of LBP: lifetime prevalence, period prevalence (last twelve months), and point prevalence (last seven days). Non-parametric tests were used to analyze the qualitative variables of this study. Fisher’s exact test was performed to compare the prevalence of LBP between sexes and age groups for variables 2 × 2. The chi-square test was used for variables 2 × k. The Mann–Whitney (or Wilcoxon–Mann–Whitney) test was used to compare the median of the difference (age) between a sample from both distributions (LBP). Backward stepwise binary logistic regression analysis was conducted with lifetime LBP prevalence as the dependent variable. The covariates used were: age, sex, educational levels, and grade. The significance level was obtained at 95% CI and *p*-value ≤ 0.05. Data analyses were carried out using SPSS^®^ IBM^®^ software, ver. 26 (IBM, Armonk, NY, USA).

## 3. Results

### Level of Back Pain Prevalence

We observed a lifetime prevalence of LBP of 49.6% (*n* = 131), a period prevalence of 46.2% (*n* = 122), and a point prevalence of 18.8% (*n* = 50). Boys reported 51.4% (*n* = 68) lifetime prevalence of LBP and girls 48.1% (*n* = 63) (*p* = 0.175, Fisher’s exact test, 2-sided) (Figure 2).

According to the educational cycle, older students registered a higher lifetime prevalence of LBP (3rd cycle: 61.8%, *n* = 81) than the younger (second cycle: 34.4%, *n* = 45; first cycle: 3.8%, *n* = 5) students (*p* < 0.001). By age group, lifetime LBP prevalence increased with age (U = 7129.5, Z = −2.6, *p* = 0.008) (Figure 3).

Period prevalence of LBP also increased with age (U = 7041.0, Z = −2.7, *p* = 0.006) (Figure 3). No statistically significant differences were found between girls 43.9% (*n* = 61) and boys 48.8% (*n* = 61) (*p* = 0.459, Fisher’s exact test, two-sided) (Figure 4).

Likewise, point prevalence of LBP increased with age, and differences were statistically significant between groups (U = 7041.0, Z = −2.7, *p* = 0.006) (Figure 5). No significant differences were found between girls 20.1% (*n* = 28) and boys 17.6% (*n* = 22) (*p* = 0.639, Fisher’s exact test, two-sided).

Only 2.3% (*n* = 6) of the students reported LBP-associated absence from school, whereas 11.4% (*n* = 30) of students reported visiting health care professionals because of LPB. Further, physical education lessons led to LBP in 16.3% (*n* = 43) of students.

Backward stepwise binary logistic regression indicated that the students in more advanced cycles had a greater probability of suffering from LBP (*p* < 0.001; OR = 2.0; 95% CI = 1.36–2.86).

## 4. Discussion

The present study aimed to analyze the prevalence of LBP in a cohort of primary school students. The binary logistic regression results we present further strengthen the evidence that lifetime prevalence of LBP increases with age, with an odds ratio (OR) of 2.1 (95% CI= 1.36–2.86) for older age, in agreement with earlier findings [2,7,9]. For instance, in Rezapur-Shahkolai et al. [26], students (7–12 years old) reported LBP in the last month, with an OR of 3.08 (95% CI: 1.80–5.26) for older age. 

In general, primary school students between six and eleven years of age (9.4 ± 1.3 years; *n* = 264) reported a 49.6% (*n* = 131) lifetime prevalence of LBP. A systematic review with participants aged between nine and sixteen showed a lifetime prevalence of 38.98% (11.60–85.56) [27]. Conversely, other studies have reported a 23% prevalence among children of the same age range (9.7 ± 2.3; *n* = 267) [28], 20% (*n* = 30) in a cohort of children between the ages of six and twelve [29], 5–7% in a younger cohort of boys and girls aged six to nine [30], and 9.01% (*n* = 36) in a cohort of students of 9.3 ± 1.87 (age interval 6–14 years) [31].

Stratified by age, seven-year-old students reported a low lifetime prevalence of LBP (33.3%, *n* = 5), and prevalence increased by age group up to a 61.3% lifetime and 56.3% period prevalence of LBP in ten-year-old students (*n* = 49). The LBP prevalence reported by ten-year-old children was similar to that reported by a group of thirteen-year-olds in a previous study [6]. Therefore, our results further strengthened the evidence that LBP onset could start as early as the age of ten.

In the study by Taimela, Kujala, Salminen, and Viljanen [32], the reported prevalence of LBP was very low (1.1%; 95% CI, 0.2–3.1%) among seven-year-old and ten-year-old (6.0%; 95% CI, 3.9–8.7%) schoolchildren, like in Wedderkopp, Leboeuf-Yde, Andersen, Froberg, and Hansen [33] who reported less than 10% prevalence between eight and ten years of age.

According to our study, the lifetime prevalence of LPB in nine-year-old students is 50.0% (*n* = 26), whereas Gunzburg et al. [34] reported a considerably higher prevalence of 36% (*n* = 142).

In our study, 9–11-year-old students reported a 54.5% (*n* = 150) lifetime prevalence of LBP, similar to a study on 571 children in the 4th and 5th grades (9–11 years old) from Southeastern (USA), which reported a 56.4% prevalence [35].

The highest prevalence of LBP was found over lifetime (49.6%, *n* = 131), in the last year (46.2%, *n* = 122), and in the last week (18.9%, *n* = 50), respectively. In contrast to other studies carried out by Cardon’s research group, the lifetime prevalence ranged from 28% to 31% [36,37] in a sample comprising 4th- and 5th-grade elementary school children (9.7 ± 0.7, range 8.1–12). However, higher values were reported in the last seven days (76.7%) [35].

Concerning LBP associated to physical education classes, in the study by Ratliffe and Hannon [35], 27.7% of students, aged 9–11 years-old, reported a higher prevalence than in our study (15.5%; *n* = 43). Regarding medical intervention for LBP from a doctor or physiotherapist, 33% of school-age children (*n* = 33) sought care in previous studies [34], while 10.8% [*n* = 30] reported seeking care in our study.

Although differences between the sexes have been reported in the literature [9,38], the results of our study are not in line with the findings of other authors [32,34].

When it comes to high prevalence, as Maher et al. [10] have already explained, there is no need to be alarmed and seek medical intervention immediately. Acute LBP could be considered a symptom rather than a disease, and the most common form of LBP is non-specific LBP. The term is used when the pathoanatomical cause of pain cannot be determined. In most cases, it appears and disappears naturally, except when it becomes chronic [39]. Therefore, the key to a better understanding of back health is investigating its development in school-age children.

Some limitations to our study should be highlighted. Firstly, it depends on the reliance and accuracy of self-reported data on LBP, which may be especially important when young children are involved. Secondly, we used an adapted version of the Nordic questionnaire not validated for young children. Thirdly, we cannot know how parental assistance for completing the questionnaire for the youngest children and students filling the questionnaire alone affected the results. However, the agreement between questionnaires (baseline) and interviews (one year follow-up) regarding the occurrence of LBP (at least one experience) for LBP in the past year and recurrent or continuous LBP was 95%, 97%, and 90%, respectively, in a cohort of young adolescents [25]. It also could indicate that different methods are equally effective for identifying a general trend. Finally, the data must be interpreted carefully, because the sample of schools was non-randomly sampled and due to the study not being large enough.

## 5. Conclusions

In summary, our findings suggest that LBP increases with age. In addition, our results further strengthen the evidence that the onset could start at ten years of age. Future studies should address kindergarten and primary school age groups in order to support public health programs as well as health educational initiatives at schools.

## Figures and Tables

**Figure 1 healthcare-09-00270-f001:**
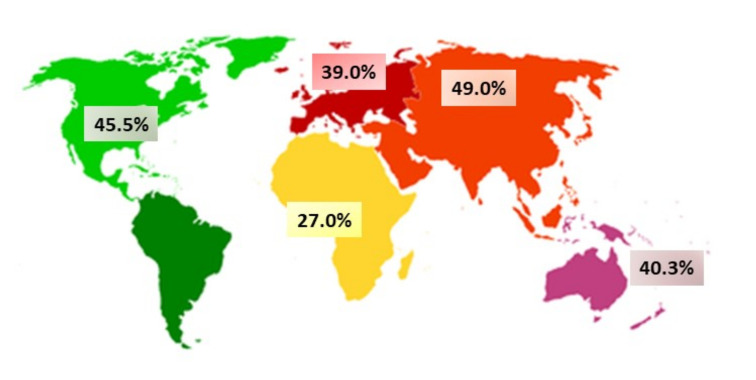
Lifetime low back pain prevalence in children and adolescent populations worldwide [2].

**Figure 2 healthcare-09-00270-f002:**
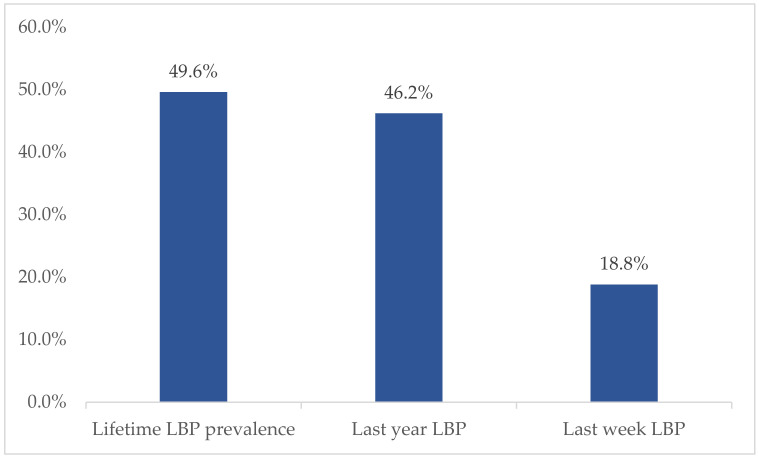
Prevalence of low back pain.

**Figure 3 healthcare-09-00270-f003:**
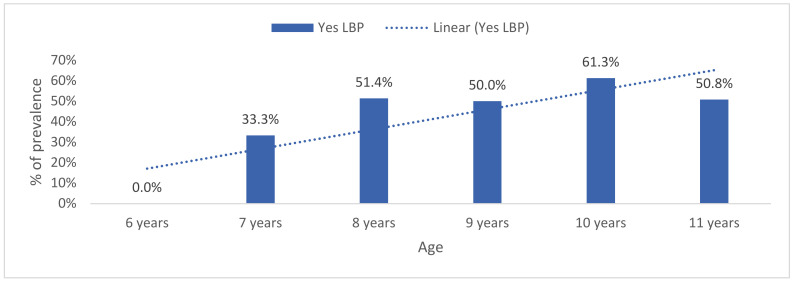
Lifetime prevalence of low back pain by age.

**Figure 4 healthcare-09-00270-f004:**
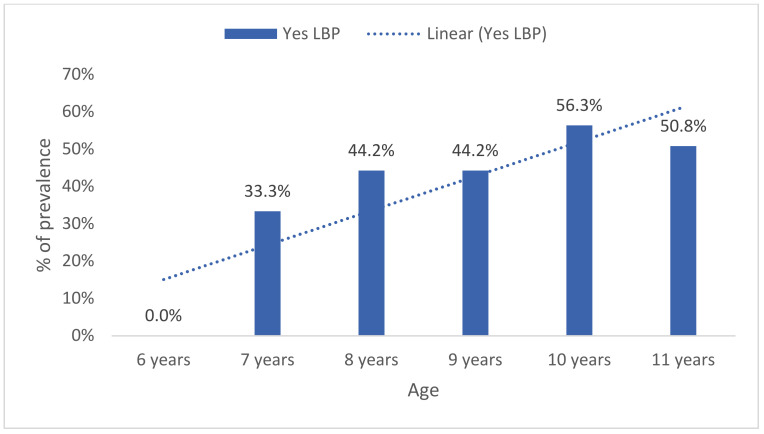
Last year’s prevalence of low back pain by age.

**Figure 5 healthcare-09-00270-f005:**
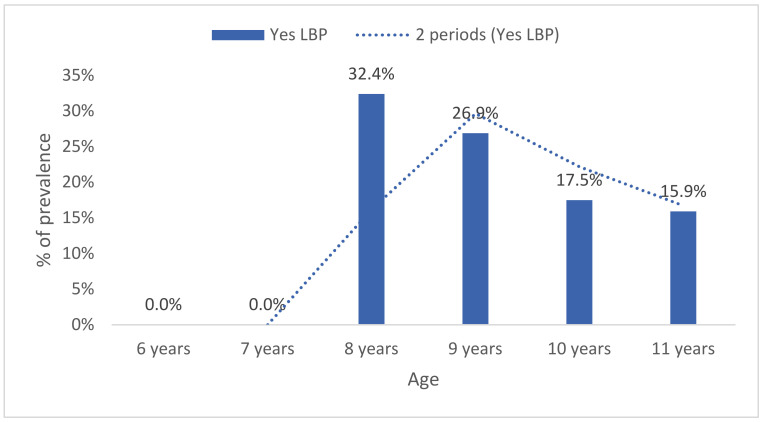
Last week’s prevalence of low back pain by age.

**Table 1 healthcare-09-00270-t001:** Descriptive data of the cohort.

Age (Years)	School 1 M ± SD	School 2 M ± SD	Total M ± SD
	9.3 ± 1.8	9.4 ± 1.2	9.4 ± 1.3
	*n* (%)	*n* (%)	*n* (%)
6	9 (10.8%)	8 (4.4%)	17 (6.4%)
7	10 (12.0%)	5 (2.8%)	15 (5.7%)
8	3 (3.6%)	34 (18.8%)	37 (14.0%)
9	13 (15.7%)	39 (21.5%)	52 (19.7%)
10	34 (41.0%)	46 (25.4%)	80 (30.3%)
11	14 (16.9%)	49 (27.1%)	63 (23.9%)
Total	97 (100%)	181 (100%)	264 (100%)

## Data Availability

The data presented in this study are available on request from the corresponding author. The data are not publicly available due to ethical restrictions.

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
