# Peer review of "Prevalence of Low Back Pain among Primary School Students from the City of Valencia (Spain)"

_healthcare, 2021, doi:10.3390/healthcare9030270_

Round 1

Reviewer 1 Report

The author performed a cross-sectional study to find the prevalence of low back pain in a sample of primary school students. The author provides very basic information. I suggest the author to perform a more statistical analysis of the data.

  • Does the author check the normality of the data?
  • For small samples use Fisher's exact test.
  • I suggest the author to perform some regression analysis to predict the prevalence of low back pain in a sample of primary school students.

Author Response

We thank the reviewers for their thoughtful and in-depth comments concerning our manuscript. Your suggestions helped us improve the quality of our paper. We carefully considered every comment and made the appropriate changes, which are highlighted in red font. Besides the manuscript has been edited by a specialist native speaker. Our point-by-point responses are also noted below.

Reviewer #1:

The author performed a cross-sectional study to find the prevalence of low back pain in a sample of primary school students. The author provides very basic information. I suggest the author to perform a more statistical analysis of the data.

  • Does the author check the normality of the data?

Author’s response: We have checked it and modified the statistical analysis and results sections.

Test results of the Kolmogorov-Smirnov data showed that the distribution is not normal (Z = .354; p < .001), so we used nonparametric tests to analyse the data. 

  • For small samples use Fisher's exact test.

Author’s response: Al the chi-square test results have been modified by Fisher’s exact test, 2-siled.

  • I suggest the author to perform some regression analysis to predict the prevalence of low back pain in a sample of primary school students.

Author’s response: It has been considered and applied:

The binary logistic regression indicated that the students belonging to the higher educational cycles had a greater probability of suffering from low back pain (p < .001; OR= 2.1; 95% CI= 1.545-2.949).

Variables en la ecuación

B

Error estándar

Wald

gl

Sig.

Exp(B)

95% C.I. para EXP(B)

Inferior

Superior

1st step

Age

,110

,106

1,079

1

,299

1,116

,907

1,373

Sex(1)

-,322

,253

1,620

1

,203

,725

,441

1,190

Educative level

-,072

,255

,079

1

,779

,931

,565

1,534

Educational cycle

,707

,521

1,837

1

,175

2,027

,730

5,631

Constante

-2,779

,674

17,017

1

,000

,062

2nd step

Age

,100

,099

1,024

1

,312

1,105

,911

1,341

Sex(1)

-,327

,252

1,675

1

,196

,721

,440

1,183

Educational cycle

,577

,242

5,670

1

,017

1,781

1,107

2,864

Constante

-2,768

,675

16,814

1

,000

,063

3rd step

Sex(1)

-,329

,252

1,707

1

,191

,719

,439

1,179

Educational cycle

,755

,165

20,937

1

,000

2,128

1,540

2,941

Constante

-2,451

,588

17,361

1

,000

,086

4th step

Educational cycle

,758

,165

21,139

1

,000

2,134

1,545

2,949

Constante

-2,632

,574

21,031

1

,000

,072

Reviewer 2 Report

The authors attempt to identity rates of low back pain (LBP) in children as young as 3 years of age across two public schools in Spain. Significant clarifications and revisions are needed in order to determine the validity of their findings, and to provide greater insight into what exactly their findings mean. 

Introduction:

1) How do the prevalence rates in Spain compare to prevalence rates of LBP in other parts of the world? 

2) LBP is a symptom rather than a disease. This is in contrast with current thinking that diagnoses chronic pain as a disease. See recent ICD-11 diagnostic criteria. Need to be specific about whether you are looking at acute or chronic LBP. 

3) The last paragraph of the introduction is a bit difficult to follow. This could be improved. 

Methods: 

1) Percentage calculation is wrong under Table 1:

(139% of girls, n= 139). 

2) Under Table 1, what is PE? Where is this defined? 

3) Not enough details on the Nordic Questionnaire. Is this a reliable and validated scale? Has it been validated to use in children as young as 3 years of age? Some of the most basic pain questionnaires (e.g. the Faces Pain Scale) are only valid in children as young as 4. This begs the question of whether the children were able to understand the questions being asked of them. If parents filled out the questionnaire for the child, was this consistently done for particular ages of the children? 

4) Were there any exclusions for children that perhaps did have a medical reason for their LBP? 

4) What is the memory of a 3-5yr old for their pain the year prior? 

Results: 

1) How much of the increasing prevalence of pain over time is just a result of increase in accuracy of reporting/memory development?

2) How representative are these 278 children, compared to other children in the population of Spain? 

3) How can you say anything about pain under the age of 6 years when your sample size consists of 14 children? Are two 5-year-olds from one school supposed to be representative of all other 5-year-olds in Spain? Presenting the data is fine, but, I think based on your sample size the most you can say that it is, is inconclusive, and spend more time discussing the results from children ages 6+, and how this data compares to previous papers - how are your findings, different/better?  

Discussion:

1) Need more insight into why there were differences between your findings and the individuals findings of other studies reported in the discussion. Currently, this discussion is rather uninteresting to read. It's not enough to have a blanket statement saying, "Assessments of LBP prevalence in children vary widely between studies depending on the age of study participants and methodological differences between studies, particularly in terms of how LBP is defined (Jones & Macfarlane, 2005)." 

2) Please expand on this justification, "However, the agreement between questionnaires and interviews regarding the occurrence of LBP (at least one experience), LBP in the past 12 months, and recurrent or continuous LBP in other studies was 95%, 97%, and 90%, respectively. What ages was this true for? 

3) Why do you think the prevalence of LBP starts to increase around 10yrs of age? 

Author Response

We thank the reviewers for their thoughtful and in-depth comments concerning our manuscript. Your suggestions helped us improve the quality of our paper. We carefully considered every comment and made the appropriate changes, which are highlighted in red font. Besides the manuscript has been edited by a specialist native speaker. Our point-by-point responses are also noted below.

Reviewer #2:

The authors attempt to identity rates of low back pain (LBP) in children as young as 3 years of age across two public schools in Spain. Significant clarifications and revisions are needed in order to determine the validity of their findings, and to provide greater insight into what exactly their findings mean. 

Introduction:

1) How do the prevalence rates in Spain compare to prevalence rates of LBP in other parts of the world? 

Author’s response: It has been considered and applied in the text as well as an Appendix (Figure 1):

Moreover, LBP is an important health problem in developed countries and in the rest of the world (Balague, Mannion et al. 2012, Hoy, March et al. 2014, Louw, Morris et al. 2007). However, literature investigating the initial onset of LBP in children between 3 and 11 years of age remains scarce (Figure 1).

Appendix B

Figure 1 Lifetime LBP prevalence in developed countries and developing ones in children and adolescent’s population (Calvo-Muñoz, Gómez-Conesa et al. 2013)

2) LBP is a symptom rather than a disease. This is in contrast with current thinking that diagnoses chronic pain as a disease. See recent ICD-11 diagnostic criteria. Need to be specific about whether you are looking at acute or chronic LBP. 

Author’s response: We are totally agreeing with you. We have improved it as you propose:

Line 37: Acute LBP is a symptom rather than a disease (Maher, Underwood et al. 2017)

3) The last paragraph of the introduction is a bit difficult to follow. This could be improved. 

Author’s response: It has been taken into account and has been improved. We hope it has been better.

Methods: 

1) Percentage calculation is wrong under Table 1:

(139% of girls, n= 139). 

Author’s response: Thank you so much for your in-depth review. It has been changed (52.7% of girls, n= 139).

2) Under Table 1, what is PE? Where is this defined? 

Author’s response: It has been changed. PE= primary education

3) Not enough details on the Nordic Questionnaire. Is this a reliable and validated scale? Has it been validated to use in children as young as 3 years of age? Some of the most basic pain questionnaires (e.g. the Faces Pain Scale) are only valid in children as young as 4. This begs the question of whether the children were able to understand the questions being asked of them. If parents filled out the questionnaire for the child, was this consistently done for particular ages of the children? 

Author’s response: We agree with your concern. We used an adapted version of the Nordic Questionnaire not validated for young children. We have improved this section. Moreover, we think that this should be clear to the reader and we have included as a limitation at the end of the discussion section.

The Faces Pain Scale (FPS; Bieri et al., Pain 41 (1990) is a self-report measure used to assess the intensity of children's pain. In our case we only wanted to report a dichotomic answer for Low Back Pain, YES or NO. So parents help their children to answer.

As we explained all the students were guided by prepared teachers and parents.

4) Were there any exclusions for children that perhaps did have a medical reason for their LBP? 

Author’s response: No students with serious problems were found. So, everyone could participate.

4) What is the memory of a 3-5yr old for their pain the year prior? 

Results: 

1) How much of the increasing prevalence of pain over time is just a result of increase in accuracy of reporting/memory development?

Author’s response: We thanks the reviewer for your concern. These questions are quite difficult to answer with precision. Also, it’s a general and a current dilemma.

It may be that LBP in children is so benign and its natural history so favourable that the memory of the episode fades away.

In our study we used several types of prevalence in order to analyze properly the symptoms with parent assistance for the youngest.

It remains unclear what is the most valid or reliable period prevalence to use for the collection of LBP prevalence rates, however, due to the stable nature of recall prevalence definition across the literature.

Memory decay is a term used to describe the gradual memory loss that occurs over time when recalling significant events (Volinn, 1997). Three factors determine the extent to which memory decay will affect the data collected on LBP prevalence: (a) the longer the time period of recall the greater the potential influence of memory decay, (b) the more significant the incident the less likely that memory decay will occur, and (c) the innate ability of the individual to recall events will influence the rate of memory decay.

The shortest period of recall is pain at the time of data collection, ie, point prevalence. However, too short a period of recall may limit the ability to collect data from sufficient subjects to develop an understanding of the risk factors associated with LBP. This is counterbalanced by the notion that collection of data on LBP reported at the time of questionnaire delivery will significantly reduce the potential for memory decay to affect the data validity.

The longest period of recall is lifetime prevalence, where the subject is asked if they have had any episode of LBP. The use of lifetime prevalence will negate the influence of forward telescoping, however memory decay presents a significant influence.

2) How representative are these 278 children, compared to other children in the population of Spain? 

Author’s response: It has been clarified and we have eliminated the kindergarten and poor data..

The study was performed in the city of Valencia (Spain), which had a total population of 40890 primary education students during the 2019-2020 school year. This age group was selected because non-specific LBP onset prevalence starts in 10–14 years old according to the literature (6, 8) and we wanted to describe what happen before this age group. The initial sample consisted of a total of 566 students sampled by convenience from two public primary schools (school 1 and school 2) in the city of Valencia, Spain. Of the invited students, 264 (46.6% recruited; 9.4±1.3 years old; 52.7% girls, n= 139) children completed the questionnaires (Table 1) and were included in the present study.

TABLE 1

The sample was comprised of a representative primary education student with a 95% confidence level and a margin of error of ±6%. In regards to the educational cycles, we observed that 32 (12.1%) belonged to the 1st cycle of primary education, 89 (33.7%) to the 2nd cycle, and 143 (54.2%) belonged to the 3rd cycle.

3) How can you say anything about pain under the age of 6 years when your sample size consists of 14 children? Are two 5-year-olds from one school supposed to be representative of all other 5-year-olds in Spain? Presenting the data is fine, but, I think based on your sample size the most you can say that it is, is inconclusive, and spend more time discussing the results from children ages 6+, and how this data compares to previous papers - how are your findings, different/better?  

Author’s response: Totally agree. Our data from kindergarten are poor and we have will focus in the representative sample of primary education students in the city of Valencia.

Discussion:

1) Need more insight into why there were differences between your findings and the individuals findings of other studies reported in the discussion. Currently, this discussion is rather uninteresting to read. It's not enough to have a blanket statement saying, "Assessments of LBP prevalence in children vary widely between studies depending on the age of study participants and methodological differences between studies, particularly in terms of how LBP is defined (Jones & Macfarlane, 2005)." 

Author’s response: We have been cited more current studies and we have started the discussion with one of our main results, the binary logistic regression.

2) Please expand on this justification, "However, the agreement between questionnaires and interviews regarding the occurrence of LBP (at least one experience), LBP in the past 12 months, and recurrent or continuous LBP in other studies was 95%, 97%, and 90%, respectively. What ages was this true for? 

Author’s response:

3) Why do you think the prevalence of LBP starts to increase around 10yrs of age? 

Author’s response:

Our work points an increase in the prevalence of nonspecific low back pain at early age, in the same way as previous studies (Burton, 1996, Jeffries, 2007, Yao, 2011,). However, there seems to be no agreement in determining the age at which this pronounced increase occurs. Yao (2011) discuss that the differences between studies could be related with the use of different instruments, lack of consensus in LPB definition or methodologic difference. Although these questions must be solved, we think that the range age of population study is another reason to consider. Studies in childhood age have traditionally been carried out at ages above 10 years-old (Yao, 2011, Murphy, 2005, Olsen, 1992, Harreby, 1999), nevertheless, our study demonstrates the importance of addressing this problem at earlier ages. Our results point to 10 years-old as the critical age.  We believe that this could be due to changes in lifestyle habits in ages prior to 10 years. Although the life habits of school children may differ between cultures, it is common for the lives of children between the ages of 6 and 8 to undergo a major change due to the beginning of a more academic schooling period. This means spending more hours sitting, reducing the time of physical activity and having to carry more weight in the backpack. From our point of view, this result indicates that we must deepen our knowledge of the schoolchildren back health at earlier age.     

Yao, Weiguang MD*; Mai, Xiaodan MD†; Luo, Chenling MD‡; Ai, Fuzhi MD§; Chen, Qing MD† A Cross-Sectional Survey of Nonspecific Low Back Pain Among 2083 Schoolchildren in China, Spine: October 15, 2011 - Volume 36 - Issue 22 - p 1885-1890 doi: 10.1097/BRS.0b013e3181faadea

Jeffries LJ , Milanese SF , Grimmer-Somers KA . Epidemiology of adolescent spinal pain: a systematic overview of the research literature. Spine 2007; 32 : 2630 – 7

Murphy S , Buckle P , Stubbs D . A cross-sectional study if self-reported back and neck pain among English schoolchildren and associated physical and psychological risk factors. Appl Ergon, 2005; 38 : 797 – 804.

Olsen TL , Anderson RL , Dearwater SR , et al. The epidemiology of low-back pain in an adolescent population . Am J Public Health 1992 ; 82 : 606 – 8.

Harreby M , Nygaard B , Jessen T , et al. Risk factors for low back pain in a cohort of 1389 Danish school children: an epi

Round 2

Reviewer 1 Report

The manuscript has been improved in the current revision and the authors addressed my previous concerns. 

Author Response

We thank the reviewer for their comments concerning our manuscript and to help us to improve the article.

Sincerely,

Authors

Reviewer 2 Report

I am very impressed by the response from the authors and the improvements made to this manuscript. It is much easier to read and understand, and the methodology is more sound with the exception of the Nordic scale not being validated for use in young children. However, this has been pointed out as a limitation of the study. 

Minor revisions are still needed: 

1) Figure 1 should be in reference to the sentence prior as it does not reflect that the data of LBP children between three and eleven is scarce. I would also change the language in this sentence and throughout the paper (if needed) to say data six through 11 years is scarce, as you have now removed the data in children aged 3-5 years. 

The authors did not address a question I raised previously: 

2) Please expand on this justification, "However, the agreement between questionnaires and interviews regarding the occurrence of LBP (at least one experience), LBP in the past 12 months, and recurrent or continuous LBP in other studies was 95%, 97%, and 90%, respectively. What ages was this true for? 

Author Response

We thank the reviewers for their comments concerning our manuscript. We carefully considered every comment and made the appropriate changes, which are highlighted in red font. Besides the manuscript has been edited by a specialist native speaker. Our point-by-point responses are also noted below.

Reviewer #2:

I am very impressed by the response from the authors and the improvements made to this manuscript. It is much easier to read and understand, and the methodology is more sound with the exception of the Nordic scale not being validated for use in young children. However, this has been pointed out as a limitation of the study. 

Minor revisions are still needed: 

  • Figure 1 should be in reference to the sentence prior as it does not reflect that the data of LBP children between three and eleven is scarce. I would also change the language in this sentence and throughout the paper (if needed) to say data six through 11 years is scarce, as you have now removed the data in children aged 3-5 years.

Author’s response: It has been changed

The authors did not address a question I raised previously: 

  • Please expand on this justification, "However, the agreement between questionnaires and interviews regarding the occurrence of LBP (at least one experience), LBP in the past 12 months, and recurrent or continuous LBP in other studies was 95%, 97%, and 90%, respectively. What ages was this true for? 

Author’s response: It has been completed. It was read in the article by Salminen et al 1995

The measurement of the students' perception of low back pain evaluated at baseline through the self-registered questionnaire and at the one year follow-up through an interview coincided in 90%.

Salminen, J. J., Erkintalo, M., Laine, M., & Pentti, J. (1995). Low back pain in the young. A prospective three-year follow-up study of subjects with and without low back pain. Spine20(19), 2101-7.

They were students of eighth grade of Turku, Finland. It is 14-15 years old. Young adolescents.

However, the agreement between questionnaires (baseline) and interviews (one year follow-up) regarding the occurrence of LBP (at least one experience), for LBP in the past year and recurrent or continuous LBP was 95%, 97%, and 90%, respectively in a cohort of young adolescents (25). It also could indicate that different methods are equally effective to identify a general trend.